# Analysis of the clinical and radiological outcomes of percutaneous cervical nucleoplasty: A case–control study

Chien-Hua Chen[1], You-Pen Chiu[2,3], Hui-Ru Ji[2,3], Chin-Ying Wu[4,5], Jeng-Hung Guo[3,6,7], Fu-Shan Jaw[1], Cheng-Di Chiu [ID][2,3,6,7,8]*

1 Department of Biomedical Engineering, National Taiwan University, Taipei, Taiwan, 2 School of Medicine, China Medical University, Taichung, Taiwan, 3 Graduate Institute of Biomedical Science, China Medical University, Taichung, Taiwan, 4 Department of Neurosurgery, China Medical University Hsinchu Hospital, Hsinchu, Taiwan, 5 Graduate Institute of Integrated Medicine, China Medical University, Taichung, Taiwan, 6 Spine Center, China Medical University Hospital, Taichung, Taiwan, 7 Department of Neurosurgery, China Medical University Hospital, Taichung, Taiwan, 8 Graduate Institute of Medical Sciences, National Defense Medical Center, Taipei, Taiwan

* cdchiu4046@gmail.com

**Data Availability Statement:** All relevant data are within the paper and its Supporting Information files.

## Abstract

### Background

Percutaneous cervical nucleoplasty (PCN) is a simple, safe, and effective treatment for contained cervical herniated intervertebral disc (CHIVD). However, few studies have compared the actual benefits of PCN against conservative treatment (CT), either clinically or radiographically.

### Purpose

The present study sought to analyze and to compare the outcomes of symptomatic contained CHIVD treated with PCN or CT.

### Methods

The present study was designed as a case–control comparative study. Patients who indicated for PCN after a failed CT for more than 6 months were recruited. After the exclusion of some patients who did not meet the selection criteria of the study, we finally enrolled 71 patients treated with PCN. In addition, another 21 patients who indicated for PCN but finally chose to receive CT continuously were also enrolled and categorized as the control group. All patients completed the 6-month follow-up. Pain levels and functional outcomes were evaluated pre- and post-operatively by assessing the visual analog scale (VAS), Oswestry Disability Index (ODI), and Neck Disability Index (NDI). Radiographic images of 72 of 104 intervened segments were collected to measure disc height and other cervical spinal alignments, such as range of motion, C2–7 Cobb's angle, and C2–7 sagittal vertical axis.

**Funding:** CDC was supported by China Medical University Hospital [DMR-111-097] for providing technical and financial support https://www.cmuh.cmu.edu.tw/Home/CmuhIndex_EN. The funders had no role in study design, data collection and analysis, decision to publish, or preparation of the manuscript.

**Competing interests:** The authors have declared that no competing interests exist.

## Results

Compared with the CT group, the PCN group showed significantly better outcomes on VAS, ODI, and NDI at the 1-month post-operative follow-up, which continued through at least the 6-month follow-up ($P < 0.01$ for VAS and $P < 0.05$ for ODI and NDI). The mean disc height significantly decreased, from $6.04 \pm 0.85$ mm to $5.76 \pm 1.02$ mm, 3 months after PCN treatment ($P = 0.003$). However, the degree of disc height decrease did not correlate with the changes of the substantial VAS improvement.

## Conclusions

To provide therapeutic benefits for symptomatic contained CHIVD patients after an invalid CT for 6 months, PCN seems to be a better option than CT. The reduced disc heights did not alter the clinical outcomes of PCN.

## Introduction

Cervical disc disorders represent a major cause of daily life difficulties among adults. On average, 1 in every 1,000 people suffers from cervical disc-related pain, which is associated with high medical expenses [1]. Cervical discogenic pain is often caused by a cervical herniated intervertebral disc (CHIVD). The size and location of the herniated disc and the associated compression imposed on adjacent nervous system structures can cause many symptoms, including pain, soreness, and numbness in the neck, shoulders, and arms [2]. Most patients with symptomatic CHIVD initially receive conservative treatment (CT), such as medication, rehabilitation, and cervical traction. Based on predictive factors of CT, if no subjective or objective improvement is observed within 6 months after CT treatment or the disease is aggravated, surgical treatment may need to be considered [3].

Anterior cervical decompression and fusion (ACDF) or artificial disc replacement (ADR), which both aim to decompress the cervical spinal cord and nerve roots, remain the gold standards of surgical treatment for CHIVD. However, these treatments are also associated with the risk of damage to nearby tissues (for example, the esophagus, recurrent laryngeal nerves, and thyroid arteries) or adjacent disc levels [4, 5]. Percutaneous cervical nucleoplasty (PCN) is a method for percutaneous disc decompression that was first permitted by the U.S. Food and Drug Administration in 1999 to treat symptomatic contained discs [6]. PCN combines nucleus pulposus ablation with thermal coagulation and has, thus, been referred to as a coblation technique [7]. PCN is effective due to the physical reduction of intradiscal pressure; a small reduction in contained disc volume results in a large fall in intradiscal pressure [8]. In addition, thermal PCN coblation therapy can further decrease pain from the chemical reduction of local inflammatory factors such as phospholipase A2 and interleukin-1 and the induction of regenerative cytokines such as interleukin-8 [9–11]. This technique is easy to perform and can effectively reduce the compression of intervertebral discs, minimizing the impacts on surrounding healthy tissues [12]. Many studies have reported PCN to be a safe and effective surgery over the long term [7, 13, 14]. Although other percutaneous techniques, such as percutaneous cervical discectomy (PCD), have been adopted by some physicians, the clinical results of PCN have not been reported to be inferior to those of PCD in several comparative studies [13, 15, 16]. To our knowledge, there is a limited number of studies that have evaluated the real benefits gained from PCN compared with CT, especially among these patients who are already indicated for

surgery [17]. Although a few studies have reported serial imaging changes after PCN, these studies have not provided and examined the factors affecting clinical outcomes [18].

To date, the role played by PCN has remained debatable, and a comparative case-control study may be necessary to determine the benefits of PCN relative to other therapeutic methods [11]. Thus, the aim of the study was to evaluate the clinical outcomes and imaging parameters in patients treated with PCN or CT to provide a better understanding of the therapeutic effects of PCN in patients suffering from symptomatic contained CHIVD.

## Materials and methods

### Patient enrollment and selection

The protocol of the present study received approval from the research committee of China Medical University Hospital (CMUH110-REC2-113, **S1 Appendix**). All methods were performed in accordance with relevant regulations. We collect data from patients who met the selection criteria for receiving PCN at a single medical center. The surgeries were performed by the same neurosurgeon. The inclusion criteria were as follows: (1) aged 20–60 years; (2) diagnosed with contained CHIVD demonstrated by magnetic resonance imaging (MRI); (3) persistent neck pain more than radicular pain without signs of improvement after six months of conservative treatment (physical therapy or medication); and (4) a limited degree of disc degeneration (>50% of disc height relative to adjacent disc). (5) Neither sensory nor motor neurologic deficit existed. Subsequently, some cases were excluded due to a history of other surgical treatment in the neck, too many painful discs (>3 segments), a combined surgery with PCN, or the patient's refusal to participate. Eventually, we collected 71 cases with complete data for the 6-month follow-up period. Twenty-one patients who met the same selection criteria but finally chose to receive CT continuously were categorized as the control group. The participants were enrolled in each group at the physician's discretion and not specifically for this study's purpose. All patients provided written informed consent and were provided with a comprehensive explanation of the study content. In order to protect the subject's privacy, the alternative serial number will be used for de-identification to improve the protection of subject's privacy and data security.

### The technique of PCN

The patient was placed in a supine position under local anesthesia with his or her neck mildly extended. With the aid of three-dimensional C-arm image and the experienced hands of the surgeon, a 17-gauge Crawford spinal needle was advanced through the avascular space between the right carotid sheath and esophagus, pointing to the nucleus pulpous of the targeted disc, and the final locations were confirmed by provocation discography with 0.2 c.c. contrast media injected into the index disc space. Then, the coblation instrument (Arthro-Care®; ArthroCare Cooperation, Sunnyvale, CA) with a 1-mm diameter was used to produce radiofrequency energy in low-temperature plasma (typically 40–70˚C) to remove a section of the compressed nucleus pulpous tissue. Three circular coblations were performed at the 1/4, 2/4 and 3/4 locations within the full length of the disc space.

### Pain and functional indexes

All enrolled patients first visited an outpatient clinic, where they were asked to indicate their pain level using a 10-cm visual analog scale (VAS) [19]. With the assistance of a clinical study nurse, they also completed the Oswestry Disability Index (ODI) and the Neck Disability Index (NDI), which measure both disability and quality of life [20, 21]. The outcome records at the

time the study subjects were categorized as either PCN or CT group prior to treatment were defined as baseline. All questionnaires were conducted at baseline, and 1, 3, and 6 months after treatments. Some surveys were completed using telephone interviews due to traveling difficulties. To evaluate the efficacy of PCN in or study subjects, the primary outcomes of the study were the questionnaire results of VAS, ODI, and NDI. The secondary outcomes were the changes of radiographic parameters after PCN.

### Radiographic parameters

MRI and X-ray imaging at baseline and 3-month follow-up were obtained to calculate the image parameters, which represent the biomechanical status of the treated spinal segments [22]. For single-level examinations, disc height and range of motion (ROM) were measured using a sagittal or flexion/extension X-ray images, respectively. The C2–7 sagittal vertical axis (SVA), which represents the distance between the plumb line from the center of the C2 to the upper-posterior edge of the C7, indicated the balance of the sagittal plane. The curvature of the cervical spine was represented by the Cobb's angle for C2–7 or the angle between the superior and inferior endplate that were employed to evaluate the cervical spinal lordosis. Both the SVA and Cobb's angle were measured according to the X-ray images. All the measurements were performed by a radiologist or neurosurgeon using RadiAnt DICOM Viewer (Medixant, Poland).

### Statistics

A two-tailed Student's *t*-test was employed to analyze differences in demographic data and image parameters across groups. The analyses of VAS, ODI, and NDI were performed using two-way analysis of variance, with between-group analyses performed using Sidak's multiple comparisons test. All results with a *P*-value less than 0.05 were deemed significant. Statistical analyses were performed using GraphPad Prism 7 (GraphPad Software, Inc., CA, US).

### Results

We enrolled 71 PCN cases and 21 CT cases who completed the 6-month follow-up assessments, including outcome evaluations and image acquisition (**Fig 1**). For the PCN group, the re-surgery rate for the same or adjacent discs was approximately 4.44% (4/90), with surgery occurring within a mean of 30.44 ± 24.36 months, including three patients who received additional PCN and one microdiscectomy with ADR. The demographic characteristics of both groups are shown in **Table 1**. The average ages were 46.18 ± 8.01 years for the PCN group and 45.34 ± 7.84 years for the CT group, and most cases were younger than 55 years, indicating a majority of younger age brackets for the study cohort. A slightly lower body mass index (BMI) was observed in the PCN group (*P* = 0.034). However, no significant differences were identified for age, baseline VAS, ODI, or NDI values between groups. After treatment, pain and functional outcome assessments (**Fig 2**) revealed significant improvements in the VAS, ODI, and NDI values for the PCN group from 1 to 6 months compared to that at baseline (*P* < 0.01 for VAS, ODI, and NDI). At 3-month follow up, in the CT group, further recoveries were also detected (*P* < 0.05 at 3 m and 6 m for VAS and at 6 m for ODI and NDI). Moreover, at the 6-month follow-up period used to assess therapeutic effects, PCN treatment was significantly superior to CT for both pain relief and functional improvements (*P* < 0.01 for VAS, *P* < 0.05 for ODI at 1 m, and for NDI at 1m and 3m). In addition, improvements in pain were not associated with demographic factors, such as age and BMI (**Fig 3**). **Table 2** shows the estimated parameters measured from the images obtained at baseline for the CT group and the pre- and post-operative periods for the PCN group, which represent the biomechanical status of the

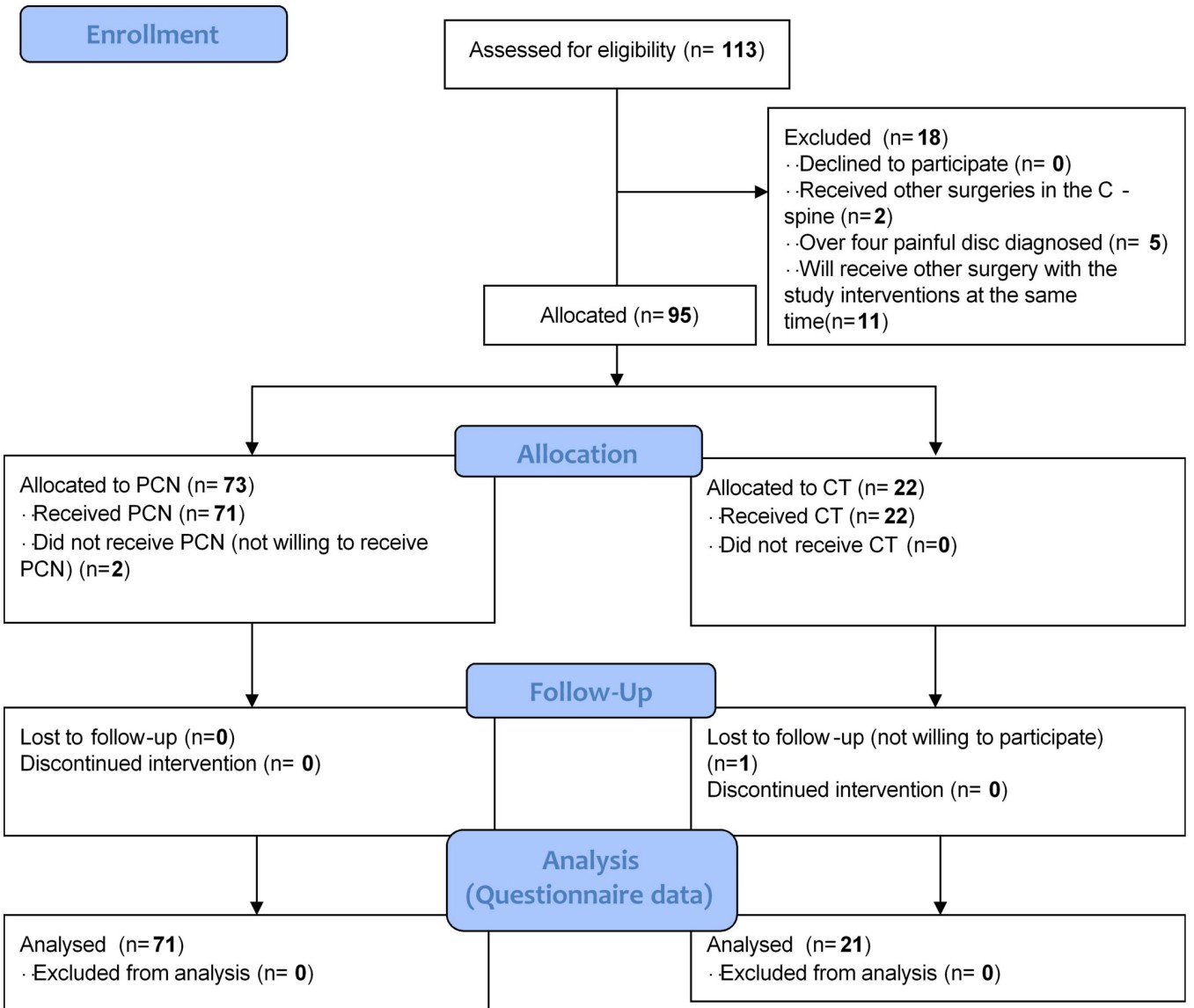

**Fig 1. A flow diagram showing patient enrollment and the eligibility criteria applied to the study.** PCN: percutaneous cervical nucleoplasty; CT: conservative treatment.

cervical spine after treatment. Of 93 measured discs treated with PCN, a significant reduction in disc height (6.04 ± 0.85 mm vs. 5.76 ± 1.02 mm) was noted as early as 3 months ($P < 0.01$). However, no correlation was observed between the loss of disc height and substantial VAS improvements (**Fig 3**). No significant within-group or between-group differences were found on the C2–7 Cobb's angle, C2–7 SVA, or ROM after treatment. These results implied that the loss of disc content was the only radiographic change observed, and neck mobility and curvature were not affected by PCN.

## Discussion

Many reports, including retrospective, prospective, meta-analyses and randomized controlled studies, have indicated that PCN is a long-term safe and effective modality for treating

**Table 1. Demographic data.**

| Study group | CT (n = 21) [a] | PCN (n = 71) | P-value |
|---|---|---|---|
| Age (y) | 43.67 ± 9.88 | 46.18 ± 8.01 | .234 |
| > 55 | 3 | 10 | |
| ≤ 55 | 18 | 61 | |
| Sex (F/M) | 9/12 | 34/38 | |
| BMI (kg/m$^2$) | 26.81 ± 4.19 | 24.23 ± 3.89 | .034 |
| Levels treated (l level/2 levels/3 levels) [b] | 10/8/3 | 22/42/7 | |
| Total disc spaces (segments) | 35 | 127 | |
| C3–4 | 6 | 14 | |
| C4–5 | 7 | 31 | |
| C5–6 | 13 | 57 | |
| C6–7 | 9 | 25 | |
| Baseline VAS (score) | 6.05 ± 1.29 | 6.51 ± 1.75 | .262 |
| Baseline ODI (%) | 34.67 ± 15.33 | 35.97 ± 20.63 | .790 |
| Baseline NDI (%) | 37.05 ± 14.77 | 41.20 ± 18.32 | .345 |

a Data is presented as the mean ± SD.

b The level treated indicates the total counts of cervical spinal segments in patients.

BMI: body mass index.

VAS: visual analog scale.

ODI: Oswestry Disability Index.

NDI: Neck Disability Index.

contained CHIVD [7, 16, 17, 23]. The effectiveness of PCN has been reported as not inferior to anterior cervical discectomy (ACD) or PCD [13, 15, 16]. Furthermore, a randomized controlled trial by Rooij et al. also suggested that for single-level contained CHIVD, long-term effectiveness is not significantly different between ACD and PCN, and concluded that PCN can be a good alternative to ACD [24]. However, Epstein et al. reported that CT could achieve similar improvements as PCN [25]. Our study found that the PCN group had significantly superior outcomes compared with the CT. group, including earlier recovery from pain and functional limitations for as long as 6 months (**Fig 2**). Unlike lumbar HIVD, less spontaneous regression was identified with CHIVD, reducing the effectiveness of CT for cervical discogenic pain [26, 27]. Our results also indicated that the extension of the CT period for more than 6 months can only gain limited effects and is less efficient than receiving PCN for patients suffering from symptomatic contained CHIVD. These findings suggested that early PCN intervention after 6 months' invalid CT should be considered to achieve prompt and long-term satisfactory results. On the other hand, with regard to safety, only some rare complications, such as inferior thyroid artery injury or spondylodiskitis, have been described [28, 29].

Cervical disc degeneration and reduced disc height are potential risk factors for the prediction of neck pain in patients [30]. However, some studies have reported that larger or optimal interbody implants do not achieve better clinical outcomes in the case of increased disc height [31, 32]. In addition, the acceleration of disc degeneration has been reported as a potential drawback of PCN [18, 33]. The present study attempted to identify temporal evidence of imaging changes after PCN. We noted a significant decrease in cervical disc height among PCN patients that could be detected as early as 3 months after surgery (**Table 2**). But no correlation was detected between changes in disc height and PCN outcomes (**Fig 3**), indicating that decreased disc height after PCN cannot be used as an indicator of poor outcome. At our center, we performed the standard three-point circular coblation technique for PCN (at 1/4, 2/4,

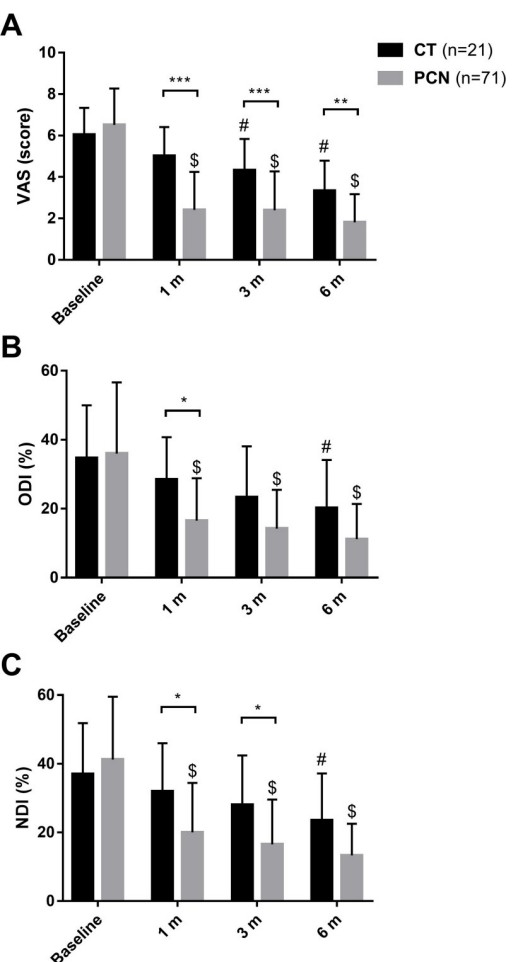

**Fig 2. Six-month follow-up of pain scales and disability indexes.** A chart showing the estimated VAS (A), ODI (B), and NDI (C) from the baseline record compared with those for the 6-month follow-up. *, **, and *** indicate between-group significance differences, with *P*-values < 0.05, < 0.01, and < 0.001, respectively. $ represents *P* < 0.01 compared with baseline value for the PCN group. # represents *P* < 0.05 compared with baseline value for the CT group. All data are presented as the mean ± SD. VAS: visual analog scale; ODI: Oswestry Disability Index; NDI: Neck Disability Index.

and 3/4) in each cervical disc space (**Fig 4**) [34]. We propose that some amount of decrease in disc height following PCN may represent an acceptable trade-off for apparent symptom improvements. However, whether the use of fewer circular coblation points would reduce the decrease in disk height or provide similar efficacy for pain relief and functional improvements remain unknown. Further large-scale studies and longer follow-up periods remain necessary to determine whether the observed decrease in disc height will result in symptom progression, which may contribute to the eventual necessity of ACDF or ADR. In our series, all patients who received PCN tolerated the whole course well, and no complications were reported. The imaging parameters, such as ROM, Cobb's angle, and the SVA, did not change, indicating that the PCN-induced decrease in disc height did not alter the sagittal or axial balance of the patients.

Klessinger reported data for 133 PCN patients, which showed a 19.5% re-surgery rate; 57.7% of the reoperations were performed in the first year after PCN. They concluded that PCN is a poor replacement for conventional open surgery [35]. However, in their retrospective cohort study, Rooij et al. reported that the reoperation rate of 158 patients over 41.5 months

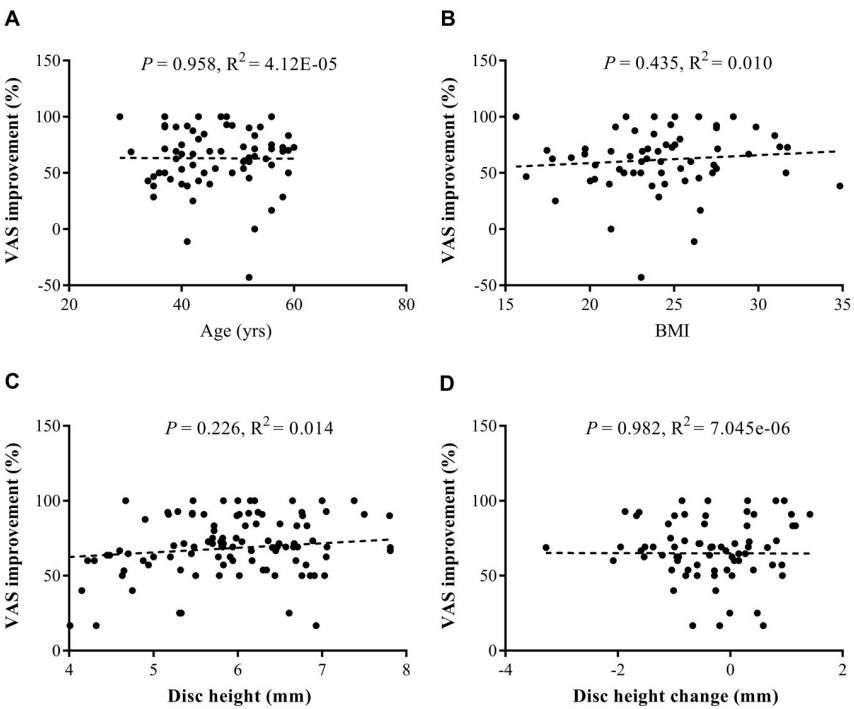

**Fig 3. Correlation analyses between substantial pain relief and prognostic factors.** The relationships between visual analog scale (VAS) improvement and age (A), body mass index (BMI) (B), baseline disc height (C), and disc height changes (D) at the 3-month follow-up were modeled by linear regression.

was 21.4%. They concluded that the reoperation rate was acceptable and suggested a potential role for PCN as a less invasive treatment option before ACD [36]. Our data show that 90 PCN patients were recruited in our series, and only one patient received ADR three months after PCN due to a lack of improvement. Another three patients received a second PCN after 11, 14, and 43 months. Our re-surgery rate was approximately 4.44% (4/90) within an average of 30.44 ± 24.36 months after surgery. This drastic difference in results between studies may be due to patient selection [34, 37]. The decision making for performing PCN should be based on several parameters which including the patients' age, symptoms, signs and the image characteristics. Despite the disc degeneration that can progress with age, the most optimal stepwise therapeutic policies should be adopted in accordance to this reason. PCN is a simple, safe, time-saving, outpatient clinical surgery that possesses good value for symptomatic contained CHIVD, especially after short-term failure to respond to CT, too early to receive open surgery, or general anesthesia are contraindicated due to complicated medical problems [13, 14]. We believe that PCN does not represent an alternative to PCD, ACD and ADR [24], but may be used to postpone or to decrease the need for eventual-step cervical fusion surgery to treat degenerative cervical disc disease [16]. However, it may have a potential role as a less invasive treatment option, before ACD is considered, for cervical radicular pain due to soft disc herniation [36].

In addition to PCN's role in alleviating cervical discogenic pain, some authors have documented its effectiveness as a treatment for cervicogenic vertigo [38, 39]. The concept of cervicogenic vertigo was first proposed by Ryan et al. in 1955 [40]. It is believed to be related to Ruffini corpuscles in degenerative cervical discs [41]. However, as definite diagnostic criteria and strong clinical evidence are lacking, the effects of PCN on cervicogenic vertigo need to be validated.

**Table 2. Estimated image parameters.**

| Image evaluations | Study group | Follow-up time (n) [a] | Statistics [b] | |
|---|---|---|---|---|
| | | | Mean ± SD | P-value |
| Disc height (mm) | CT | baseline (31) | 6.11 ± 1.19 | .217 |
| | PCN | baseline (93) | 6.04 ± 0.85 | **.003** |
| | | 3 m (93) | 5.76 ± 1.02 | |
| Cobb's angle (degrees) | CT | baseline (18) | 10.72 ± 4.82 | .808 |
| | PCN | baseline (40) | 10.13 ± 6.61 | .413 |
| | | 3 m (40) | 11.41 ± 8.10 | |
| SVA (mm) | CT | baseline (18) | −12.62 ± 6.75 | .179 |
| | PCN | baseline (40) | −9.96 ± 6.95 | .327 |
| | | 3 m (40) | −11.75 ± 10.17 | |
| ROM (degrees) | CT | baseline (22) | 4.69 ± 3.58 | .123 |
| | PCN | baseline (57) | 6.38 ± 4.58 | .717 |
| | | 3 m (57) | 6.62 ± 4.34 | |

Sagittal X-ray images was used to measure disc height, Cobb's angle, and sagittal vertical axis (SVA). Extensional and flexional X-ray images were used to measure the range of motion (ROM). All post-operative measurements were obtained from the newest follow-up images after 3 months.

[a] Some treated discs were excluded due to blurry X-ray images, incomplete image collection, or images from outside facilities. The n values for disc height and ROM represent the number of evaluated disc, whereas the n values for Cobb's angle and SVA represent total cases.

[b] Statistical analyses of between-group differences were performed using Student's t-test.

CT: conservative treatment.

PCN: percutaneous cervical nucleoplasty.

SVA: sagittal vertical axis.

ROM: range of motion.

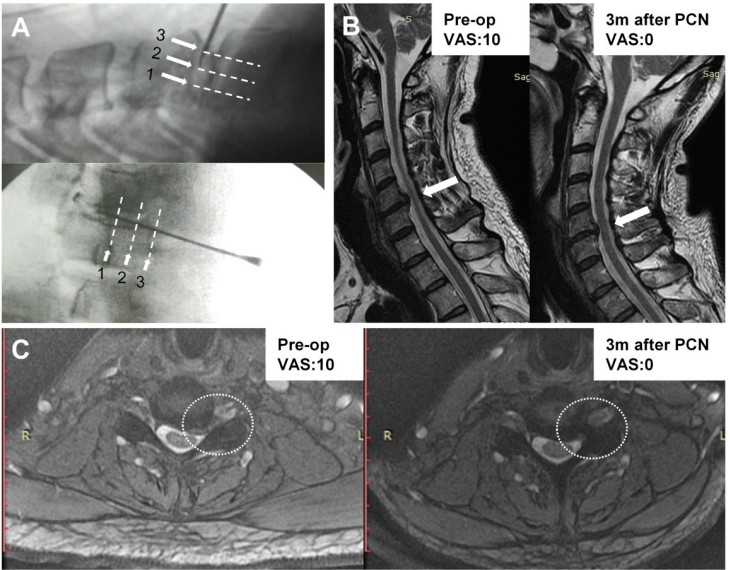

**Fig 4. Radiographic and magnetic resonance images from a sample patient.** (A) Representative intraoperative images showing the three-point circular coblation technique of PCN. The white arrows and dotted lines indicate the sequence and the target locations for the coblations performed in the target disc. Pre- and post- operative images of sagittal (B) and axial (C) T2-weighted MRI from a 48-year-old patient with C5-6 disc protrusion. The white arrows/circle indicate the improvement of the entrapped left C5/6 nerve root due to CHIVD.

Although this study conducted follow-up X-rays of the cervical spine, some patients did not come to the clinic for these final imaging analyses, resulting in loss to follow-up, which might have led to some inconsistencies in the total number of cases included in the analysis. The present study also has a number of other limitations, such as the small number of cases, lack of randomization, and use of only two-dimensional follow-up images. In addition, six months of follow-up may be insufficient. A longer follow-up period may provide additional information on the recovery of the CT group and recurrent pain after surgery. However, we believe that these preliminary data can provide a new and useful clinical understanding of the effectiveness of PCN.

## Conclusion

For contained CHIVD patients with symptoms after a period of invalid CT, prompt treatment with PCN instead of persistent CT is recommended. The progression of disc degeneration due to the loss of disc height is not a critical issue for the mid-term outcomes of PCN, which was associated with a low rate of re-surgery.

## Supporting information

**S1 Appendix. Institutional Review Board certificate of the study protocol.**
(PDF)

## Author Contributions

**Conceptualization:** Cheng-Di Chiu.

**Data curation:** Hui-Ru Ji, Chin-Ying Wu, Jeng-Hung Guo.

**Formal analysis:** You-Pen Chiu.

**Funding acquisition:** Cheng-Di Chiu.

**Methodology:** Chin-Ying Wu, Jeng-Hung Guo.

**Project administration:** Chien-Hua Chen, You-Pen Chiu.

**Supervision:** Cheng-Di Chiu.

**Writing – original draft:** Chien-Hua Chen, You-Pen Chiu.

**Writing – review & editing:** Fu-Shan Jaw, Cheng-Di Chiu.

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
