## [Decision Letter · Decision Letter 0]

10 Oct 2022

PONE-D-22-22166Analysis of the Clinical and Radiological Outcomes of Percutaneous Cervical Nucleoplasty: A Case–Control StudyPLOS ONE

Dear Dr. Chiu,

Thank you for submitting your manuscript to PLOS ONE. After careful consideration, we feel that it has merit but does not fully meet PLOS ONE’s publication criteria as it currently stands. Therefore, we invite you to submit a revised version of the manuscript that addresses the points raised during the review process.

We look forward to receiving your revised manuscript.

Kind regards,

Andrea Giannini

Academic Editor

PLOS ONE

Journal Requirements:

2. During the internal evaluation of the documents provided. It is in our understanding that the participants were enrolled into each surgical group independent of research, at the physician's discretion. As such we would be grateful if you could report this within the mansucript text to avoid confusion with a possible clinical trial. Please also remove the TREND check list and IRB approved study protocols from the supporting material as this is not required for your article type.

   "Dr. Cheng-Di Chiu was supported by China Medical University Hospital [DMR-111-097] for providing technical and financial support."

   "CDC was supported by China Medical University Hospital [DMR-111-097] for providing technical and financial support https://www.cmuh.cmu.edu.tw/Home/CmuhIndex_EN.

Additional Editor Comments:

Dear authors,

the topic of the present article titled “Analysis of the Clinical and Radiological Outcomes of Percutaneous Cervical Nucleoplasty: A Case–Control Study” is very interesting, the paper and the aim falls within the scope of the journal but the article needs major improvements.

Review some spellings. Methods and discussion should be modified and improved.

I suggest improving the manuscript with the reviewers' comments.

Reviewers' comments:

Reviewer's Responses to Questions

**Comments to the Author**

1. Is the manuscript technically sound, and do the data support the conclusions?

Reviewer #1: Partly

Reviewer #2: Yes

Reviewer #3: Yes

Reviewer #4: Yes

2. Has the statistical analysis been performed appropriately and rigorously? 

Reviewer #1: Yes

Reviewer #2: Yes

Reviewer #3: Yes

Reviewer #4: Yes

3. Have the authors made all data underlying the findings in their manuscript fully available?

Reviewer #1: Yes

Reviewer #2: Yes

Reviewer #3: Yes

Reviewer #4: Yes

4. Is the manuscript presented in an intelligible fashion and written in standard English?

Reviewer #1: Yes

Reviewer #2: Yes

Reviewer #3: Yes

Reviewer #4: Yes

5. Review Comments to the Author

Reviewer #1: This is my review about "Analysis of the Clinical and Radiological Outcomes of Percutaneous Cervical Nucleoplasty: A Case–Control Study".

The study deals with the outcomes of symptomatic 34

contained CHIVD treated with PCN or CT.

The present study was designed as a case–control comparative study. Patients who

indicated for PCN after a failed CT for more than 6 months were recruited.

Could you please comment to these more recent papers and compare with your results:

2022 May 17;15:1433-1441. doi: 10.2147/JPR.S359512. eCollection 2022.

Long-Term Clinical Results of Percutaneous Cervical Nucleoplasty for Cervical Radicular Pain: A Retrospective Cohort Study

Judith Divera de Rooij

2020 Nov;23(6):553-564.

The Effect of Percutaneous Nucleoplasty vs Anterior Discectomy in Patients with Cervical Radicular Pain due to a Single-Level Contained Soft-Disc Herniation: A Randomized Controlled Trial

Judith de Rooij

2020 Jan;133:e205-e210. doi: 10.1016/j.wneu.2019.08.210. Epub 2019 Sep 5.

Long-Term Clinical Outcomes of Percutaneous Cervical Nucleoplasty for Cervical Degenerative Diseases with Neck Pain and Cervical Vertigo

Chungen Li

2019 May;22(3):E205-E214.

Therapeutic Effects and Safety of Percutaneous Disc Decompression with Coblation Nucleoplasty in Cervical Vertigo: A Retrospective Outcome Study with 74 Consecutive Patients and Minimum 1-Year Follow-Up

Shangfu Li

Reviewer #2: The manuscript technically sound,the data support the data and conclusion ,the statistical analysis been performed appropriately and rigorously

The manuscript presented in an intelligible fashion and written in standard English

Reviewer #3: The manuscript was good written and can be accepted.

English language is good.

The results were good arranged after well described materials and methods.

The results were supported by figures

Discussion was good including conclusions

Reviewer #4: First of all, I apologize you to say rude things.

This manuscript seems a kind of supplementary examination. Many previous manuscripts announced the effectiveness of PCN, nevertheless you tried to support it. Even the conclusion in this manuscript does not sound new and impressive.

#1. In line 78-79, you mentioned the chemical effectiveness of PCN using the literature written by Ren D et al. They had referred to the correlation between PCN and PLA2, not IL-1, IL-6 and TNF-alpha specifically. More precise information should be written in this manuscript.

#2. Compare to other literatures concerning about PCN, the follow-up period (six months) seems to be shorter in this study. Although the VAS score or other indicators drastically and rapidly decreased in PCN group, gradual but continuous decrease was shown even in CT group. Longer follow-up may lead the different conclusion.

6. PLOS authors have the option to publish the peer review history of their article (what does this mean?). If published, this will include your full peer review and any attached files.

Reviewer #1: No

Reviewer #2: No

Reviewer #3: No

Reviewer #4: No

---

## [Author Response · Author response to Decision Letter 0]

2 Nov 2022

Andrea Giannini, PhD

Academic Editor

PLOS ONE

Dr. Giannini:

We appreciate the opportunity to revise our manuscript entitled, “Analysis of the Clinical and Radiological Outcomes of Percutaneous Cervical Nucleoplasty: A Case–Control Study.” The manuscript ID is PONE-D-22-22166, authored by Chien-Hua Chen et al. Our responses to the comments from the editors and reviewers are included below.

We acknowledge the reviewers’ constructive comments and have addressed each comment in our point-by-point response. We have incorporated various corrections into the revised manuscript, according to the suggestions made by the reviewers and editors. A detailed list of our replies to the reviewers’ comments and the changes made in response to these comments is provided on the following pages of the “Authors’ Response to Editors and Reviewers.” Please feel free to contact us with any questions.

Your kind consideration of our manuscript is highly appreciated.

Sincerely yours,

Authors’ Response to Editors and Reviewers

Academic editor’s points

1. Please ensure that your manuscript meets PLOS ONE's style requirements, including those for file naming. The PLOS ONE style templates can be found at https://journals.plos.org/plosone/s/file?id=wjVg/PLOSOne_formatting_sample_main_body.pdf

Answer: The manuscript has been amended to meet PLOS ONE's style requirements.

2. During the internal evaluation of the documents provided. It is in our understanding that the participants were enrolled into each surgical group independent of research, at the physician's discretion. As such we would be grateful if you could report this within the mansucript text to avoid confusion with a possible clinical trial. Please also remove the TREND check list and IRB approved study protocols from the supporting material as this is not required for your article type.

Answer: Thank you. We have reported it in the manuscript text to avoid confusion (Lines 104–105). The unnecessary documents have been removed from the supporting materials.

3. Thank you for stating the following in the Acknowledgments Section of your manuscript: "Dr. Cheng-Di Chiu was supported by China Medical University Hospital [DMR-111-097] for providing technical and financial support." We note that you have provided funding information that is not currently declared in your Funding Statement. However, funding information should not appear in the Acknowledgments section or other areas of your manuscript. We will only publish funding information present in the Funding Statement section of the online submission form. Please remove any funding-related text from the manuscript and let us know how you would like to update your Funding Statement. Currently, your Funding Statement reads as follows: "CDC was supported by China Medical University Hospital [DMR-111-097] for providing technical and financial support https://www.cmuh.cmu.edu.tw/Home/CmuhIndex_EN. The funders had no role in study design, data collection and analysis, decision to publish, or preparation of the manuscript." Please include your amended statements within your cover letter; we will change the online submission form on your behalf.

Answer: Thank you for this reminder. The current Funding Statement in the online submission is correct so there is no need to update it. We have removed any funding-related text from the manuscript.

Answer: Thank you. The ethics statement is now only in the Methods section in the revised version of the manuscript. 

Answer: Thank you. We have added a caption and an in-text citation for the supporting information (Lines 92 and 408).

6. Additional Editor Comments: Dear authors, the topic of the present article titled “Analysis of the Clinical and Radiological Outcomes of Percutaneous Cervical Nucleoplasty: A Case–Control Study” is very interesting, the paper and the aim falls within the scope of the journal but the article needs major improvements. Review some spellings. Methods and discussion should be modified and improved. I suggest improving the manuscript with the reviewers' comments.

Answer: Thank you. We have further improved the manuscript based on your suggestions and the reviewers’ comments.

Reviewers’ points

Reviewer #1

1. The study deals with the outcomes of symptomatic contained CHIVD treated with PCN or CT. The present study was designed as a case–control comparative study. Patients who indicated for PCN after a failed CT for more than 6 months were recruited. Could you please comment to these more recent papers and compare with your results? 2022 May 17;15:1433-1441. doi: 10.2147/JPR.S359512. eCollection 2022. Long-Term Clinical Results of Percutaneous Cervical Nucleoplasty for Cervical Radicular Pain: A Retrospective Cohort Study Judith Divera de Rooij 2020 Nov;23(6):553-564. The Effect of Percutaneous Nucleoplasty vs Anterior Discectomy in Patients with Cervical Radicular Pain due to a Single-Level Contained Soft-Disc Herniation: A Randomized Controlled Trial Judith de Rooij 2020 Jan;133:e205-e210. doi: 10.1016/j.wneu.2019.08.210. Epub 2019 Sep 5. Long-Term Clinical Outcomes of Percutaneous Cervical Nucleoplasty for Cervical Degenerative Diseases with Neck Pain and Cervical Vertigo Chungen Li 2019 May;22(3):E205-E214. Therapeutic Effects and Safety of Percutaneous Disc Decompression with Coblation Nucleoplasty in Cervical Vertigo: A Retrospective Outcome Study with 74 Consecutive Patients and Minimum 1-Year Follow-Up Shangfu Li

Answer: Thank you for the comments and suggestion. We have updated the manuscript, commented on these more recent papers, and compared their results with ours (Lines 231–233, 265–270, 283–291).

Reviewer #2

1. The manuscript technically sound,the data support the data and conclusion ,the statistical analysis been performed appropriately and rigorously The manuscript presented in an intelligible fashion and written in standard English.

Answer: Thank you for your evaluation and comments.

Reviewer #3

1. The manuscript was good written and can be accepted. English language is good. The results were good arranged after well described materials and methods. The results were supported by figures. Discussion was good including conclusions.

Answer: Thank you for your evaluation and comments.

Reviewer #4

1. First of all, I apologize you to say rude things. This manuscript seems a kind of supplementary examination. Many previous manuscripts announced the effectiveness of PCN, nevertheless you tried to support it. Even the conclusion in this manuscript does not sound new and impressive. In line 78-79, you mentioned the chemical effectiveness of PCN using the literature written by Ren D et al. They had referred to the correlation between PCN and PLA2, not IL-1, IL-6 and TNF-alpha specifically. More precise information should be written in this manuscript.

Answer: Thank you for your comments. We have incorporated more precise information in the introduction, which addresses the impact of PCN on IL-1and IL-8 expression in disc tissue (Lines 68–73). In addition to its degradation of PLA2 activity, another study provided evidence that nucleoplasty decreased IL-1 expression and promoted IL-8 expression, implying the impacts of nucleoplasty on disc regeneration.

2. Compare to other literatures concerning about PCN, the follow-up period (six months) seems to be shorter in this study. Although the VAS score or other indicators drastically and rapidly decreased in PCN group, gradual but continuous decrease was shown even in CT group. Longer follow-up may lead the different conclusion.

Answer: Thank you for your comments. We agree with the reviewer’s perspective on the follow-up period. A longer follow-up period may provide additional information on the recovery of the CT group and recurrent pain after surgery. We discuss the insufficient length of the follow-up period in the section on limitations (Lines 296–298). To our knowledge, two studies with control participants had a longer follow-up period compared to the present study (ref. 11 & 23). However, our study recruited patients who had received CT for 6 months prior to enrollment. This means that the control participants in our study had already received 12 months of CT by the 6th month of the study period, at the time when PCN still showed a higher effect on pain relief than CT. We emphasize that timely PCN treatment may be a better choice for reducing pain compared to continued and lengthy CT.

---

## [Decision Letter · Decision Letter 1]

28 Nov 2022

Analysis of the Clinical and Radiological Outcomes of Percutaneous Cervical Nucleoplasty: A Case–Control Study

PONE-D-22-22166R1

Dear Dr. Cheng-Di Chiu,

We’re pleased to inform you that your manuscript has been judged scientifically suitable for publication and will be formally accepted for publication once it meets all outstanding technical requirements.

Kind regards,

Andrea Giannini

Academic Editor

PLOS ONE

Additional Editor Comments (optional):

The manuscript has been modified with the comments of the reviewers. It is now ready to be published.

Reviewers' comments:

Reviewer's Responses to Questions

**Comments to the Author**

1. If the authors have adequately addressed your comments raised in a previous round of review and you feel that this manuscript is now acceptable for publication, you may indicate that here to bypass the “Comments to the Author” section, enter your conflict of interest statement in the “Confidential to Editor” section, and submit your "Accept" recommendation.

Reviewer #1: All comments have been addressed

Reviewer #2: All comments have been addressed

Reviewer #3: All comments have been addressed

Reviewer #4: All comments have been addressed

2. Is the manuscript technically sound, and do the data support the conclusions?

Reviewer #1: Yes

Reviewer #2: Yes

Reviewer #3: Yes

Reviewer #4: Yes

3. Has the statistical analysis been performed appropriately and rigorously? 

Reviewer #1: Yes

Reviewer #2: Yes

Reviewer #3: I Don't Know

Reviewer #4: Yes

4. Have the authors made all data underlying the findings in their manuscript fully available?

Reviewer #1: Yes

Reviewer #2: Yes

Reviewer #3: Yes

Reviewer #4: Yes

5. Is the manuscript presented in an intelligible fashion and written in standard English?

Reviewer #1: Yes

Reviewer #2: Yes

Reviewer #3: Yes

Reviewer #4: (No Response)

6. Review Comments to the Author

Reviewer #1: (No Response)

Reviewer #2: interested research with appropriate language and correct statistically. The results was excited.

the research technically sound piece of scientific research with the data support the conclusions. Experiments have been conducted rigorously, with appropriate controls, replication, and sample sizes. The conclusions have be drawn appropriately based on the data presented.

Reviewer #3: The manuscript is interesting and good written and discussed.

Introduction: was good and include aim of the work.

Materials and methods: were good designed.

Results: were good described.

Discussion: was good written.

Reviewer #4: Thank you for correcting your manuscript.

In the present study, the superiority of PCN after long-term follow-up is uncertain compared with CT, however, the the potential of rapid recovery is almost obvious. Hence, this manuscript must be a boon for the patient suffering from CHIVD.

7. PLOS authors have the option to publish the peer review history of their article (what does this mean?). If published, this will include your full peer review and any attached files.

Reviewer #1: **Yes: **Jens-Christian Altenbernd

Reviewer #2: No

Reviewer #3: No

Reviewer #4: No

---

## [Editor Report · Acceptance letter]

1 Dec 2022

PONE-D-22-22166R1 

Analysis of the clinical and radiological outcomes of percutaneous cervical nucleoplasty: A case–control study 

Dear Dr. Chiu:

I'm pleased to inform you that your manuscript has been deemed suitable for publication in PLOS ONE. Congratulations! Your manuscript is now with our production department. 

Kind regards, 

on behalf of

Dr. Andrea Giannini 

Academic Editor

PLOS ONE